# Hyphae of *Rhizopus arrhizus* and *Lichtheimia corymbifera* Are More Virulent and Resistant to Antifungal Agents Than Sporangiospores In Vitro and in *Galleria mellonella*

**DOI:** 10.3390/jof9100958

**Published:** 2023-09-23

**Authors:** Rex Jeya Rajkumar Samdavid Thanapaul, Ashleigh Roberds, Kariana E. Rios, Thomas J. Walsh, Alexander G. Bobrov

**Affiliations:** 1Wound Infections Department, Bacterial Diseases Branch, Center for Infectious Diseases Research, Walter Reed Army Institute of Research, Silver Spring, MD 20910, USA; haiiamrex@gmail.com (R.J.R.S.T.); kariana.e.rios.ctr@health.mil (K.E.R.); 2NRC Research Associateship Programs, National Academies of Sciences, Engineering, and Medicine, Washington, DC 20001, USA; 3Oak Ridge Institute for Science and Education, Oak Ridge, TN 37830, USA; 4Center for Innovative Therapeutics and Diagnostics, Richmond, VA 23220, USA; tjwalsh@citdx.org; 5Department of Medicine and Microbiology & Immunology, University of Maryland School of Medicine, Baltimore, MD 21201, USA

**Keywords:** mucormycosis, *Rhizopus arrhizus*, *Lichtheimia corymbifera*, *Galleria mellonella*, virulence, hyphae, antifungal agents, efficacy, minimum inhibitory concentration assay

## Abstract

*Mucorales* species cause debilitating, life-threatening sinopulmonary diseases in immunocompromised patients and penetrating wounds in trauma victims. Common antifungal agents against mucormycosis have significant toxicity and are often ineffective. To evaluate treatments against mucormycosis, sporangiospores are typically used for in vitro assays and in pre-clinical animal models of pulmonary infections. However, in clinical cases of wound mucormycosis caused by traumatic inoculation, hyphal elements found in soil are likely the form of the inoculated organism. In this study, *Galleria mellonella* larvae were infected with either sporangiospores or hyphae of *Rhizopus arrhizus* and *Lichtheimia corymbifera*. Hyphal infections resulted in greater and more rapid larval lethality than sporangiospores, with an approximate 10–16-fold decrease in LD_50_ of hyphae for *R. arrhizus* (*p* = 0.03) and *L. corymbifera* (*p* = 0.001). Liposomal amphotericin B, 10 mg/kg, was ineffective against hyphal infection, while the same dosage was effective against infections produced by sporangiospores. Furthermore, in vitro, antifungal susceptibility studies show that minimum inhibitory concentrations of several antifungal agents against hyphae were higher when compared to those of sporangiospores. These findings support using hyphal elements of *Mucorales* species for virulence testing and antifungal drug screening in vitro and in *G. mellonella* for studies of wound mucormycosis.

## 1. Introduction

Over the past two decades, there has been an unprecedented surge in post-traumatic invasive mold infections [1,2]. Many of these cases were in wounded U.S. military personnel during the war in Afghanistan [3,4]. Fungi of the order Mucorales were the predominant cause of angio-invasive fungal infections in combat casualties [3,5,6,7,8,9,10,11]. *Apophysomyces* species (spp.), *Saksenaea* spp., *Mucor* spp., and *Lichtheimia* spp. were reported to be the predominant cause of post-traumatic mucormycosis [12,13,14,15]. *Rhizopus* spp. are responsible for more than 50% of all mucormycosis cases in the respiratory tract or central nervous system worldwide and are often found in cutaneous wounds [16,17].

While sporangiospores are assumed to be the inoculating form of the Mucorales in sinopulmonary disease, we hypothesized that hyphae found in soil and biofilm matrices on stone, wood, glass, and organic debris are responsible for infection in post-traumatic wounds [15]. Some observations in the available literature support this rationale. One such observation is that substantial quantities of sporangiospores are required to establish cutaneous infection in animal models [18,19,20,21,22,23,24,25]. Additionally, *Apophysomyces* spp. and *Saksenaea* spp. do not readily produce sporangiospores [26,27]. Therefore, we propose that mucormycosis is more readily induced by inoculation with hyphal elements rather than sporangiospores, and that this form of inoculum results in greater virulence and resistance to antifungal therapy.

Combat wound-related mucormycosis is a highly aggressive disease frequently refractory to antifungal therapy and often managed by amputation [9,10]. The therapeutic armamentarium of antifungal agents for the treatment of mucormycosis is limited to amphotericin B (AmB), isavuconazole, and posaconazole. However, these agents have a low frequency of therapeutic success in treating mucormycosis, particularly in combat-related trauma. Therefore, new therapeutic solutions are urgently needed to improve clinical outcomes of this frequently fatal and debilitating infection in both civilian and military patient populations. Preclinical models should be established to understand pathogenesis and develop new antifungal therapeutics for combat wound-related mucormycosis.

*Galleria mellonella* larvae are simple invertebrate model organisms widely used to study microbial virulence and antimicrobial activity. Low cost, rapid result generation, and absence of ethical/legal considerations make these larvae very appealing to study host-pathogen interactions [28]. Recently, *G. mellonella* larvae have been used to establish lethal infections of multiple *Mucorales* species [29]. Moreover, the *Galleria* model of mucormycosis was validated for successful treatment with standard-of-care (SOC) antifungal agents [29,30]. However, in all published reports, sporangiospores were used as the inoculum, mimicking the infection route for sinopulmonary disease [17].

In this study, we aimed to develop *Galleria* mucormycosis models using hyphal elements of *Mucorales* strains to more accurately replicate the conditions of traumatic inoculation encountered in combat-induced injury. We chose *Rhizopus arrhizus* and *Lichtheimia corymbifera* for these studies because they cause severe wound infections and readily produce large quantities of sporangiospores. Germinated sporangiospores/hyphal elements and non-germinated sporangiospores were compared for virulence and antifungal therapy, using liposomal amphotericin B (L-AmB) to treat *Galleria* mucormycosis. Finally, minimum inhibitory concentrations (MICs) of L-AmB, AmB, and posaconazole were compared against hyphae and sporangiospores in vitro. Overall, we found that hyphae of both fungal species were significantly more virulent and resistant to L-AmB than sporangiospores in larvae mucormycosis.

## 2. Materials and Methods

### 2.1. Organisms and Growth Conditions

For these studies, we used the virulent strains *R. arrhizus* and *L. corymbifera*, which were reported to be good producers of sporangiospores [31]. The isolate of *R. arrhizus* (ATCC 56536) was a reference strain isolated from a postmortem lung issue of a man with acute lymphoblastic leukemia in the Netherlands. A recent study of this strain demonstrated virulence with a median survival time of five days for mice inoculated with 10^6^ sporangiospores/immunocompetent mice [32]. *L. corymbifera* (ATCC 46771), a neotype isolated from soil in Afghanistan, is reported to be virulent in various in vitro and in vivo studies [33,34,35,36].

The glycerol stock of *R. arrhizus* was grown in Petri dishes containing Synthetic Mucor Medium Agar (SMMA) [arabinose: 4.0 g/L; asparagine: 2.0 g/L, potassium dihydrogen phosphate: 0.5 g/L, magnesium sulfate hexahydrate: 0.25 g/L, thiamine HCl: 0.5 mg/L; agar: 15 g/L). This formulation was based on a liquid medium used to obtain sporangiospores from *Saksenaea vasiformis* [37]. *L. corymbifera* was grown in petri dishes (10–12) containing Potato Dextrose Agar (PDA) (Difco™) at 37 °C for 7 days. Sporangiospores were washed off the plates with 10-to-15 mL of 0.9% sterile saline containing 0.05% of tween 80, followed by a gentle rub of sporangiospores with a sterile cell spreader until fully suspended. The harvested sporangiospores were filtered through a 70-µm filter to remove clumps of hyphal elements. The sporangiospore suspension was centrifuged at 6000 rpm for 10 min and washed twice with sterile saline. The pellet was resuspended in 5 mL of 0.9% of sterile saline solution, and the concentration of sporangiospores was calculated after counting with a hemocytometer (Nexcelom Bioscience LLC). For hyphae/germling preparation, a total of 1.0 × 10^8^–1.0 × 10^9^ *R. arrhizus* and *L. corymbifera* sporangiospores were incubated in 5–10 mL RPMI 1640 containing 10% fetal bovine serum (FBS) under constant shaking at 150 rpm at 30 °C for 24 h. Germlings were centrifuged at 6000 rpm for 10 min, washed twice with 0.9% sterile saline, and resuspended with 5 mL of 0.9% sterile saline, and the concentration of germlings was calculated by counting with a hemocytometer. Colony forming units (CFUs) were determined 1 day prior to infection by plating 100 µL of inoculum dilutions on RPMI agar plates and incubating them overnight at 30 °C. RPMI agar plates were made by reconstituting the filter-sterilized RPMI liquid medium [16.2 g/L of RPMI-1640 powder medium (Gibco^TM^) (with L-glutamine, phenol red, and without sodium bicarbonate) buffered with 35 g/L of MOPS (morpholine propane sulfonic acid) and 20 g/L of glucose] into an autoclaved agar medium containing 15 g/L of agar.

### 2.2. Optimization of Media for the Production of Sporangiospores and Generation of Hyphae of R. arrhizus and L. corymbifera

Inocula with a high concentration of sporangiospores are needed for animal models of mucormycosis. We studied several methods and multiple growth media to optimize sporangiospore production. For *R. arrhizus,* we obtained the following sporangiospore suspensions: 1.65 × 10^9^ sporangiospores/plate on SMMA, 2.0 × 10^8^ sporangiospores/plate on PDA, 8.8 × 10^5^ sporangiospores/plate on Sabouraud Dextrose Agar (SDA), and 3.9 × 10^5^ sporangiospores/plate on Yeast Peptone Dextrose Agar (YPDA). For *L. corymbifera*, we obtained the following sporangiospores suspensions: 2.86 × 10^9^ sporangiospores/plate on PDA and 2.24 × 10^9^ sporangiospores/plate on SMMA. Thus, we identified that *R. arrhizus* and *L. corymbifera* produced a maximal number of viable sporangiospores on SMMA and PDA.

To stimulate hyphal production, freshly isolated sporangiospores were incubated in RPMI-1640 medium (with glutamine and without sodium bicarbonate) containing 10% fetal bovine serum (FBS) with constant shaking (150 rpm) for 24 h at 30 °C. We observed that this procedure resulted in a mixture of developed hyphae, germinated sporangiospores, swollen sporangiospores, and original sporangiospores.

### 2.3. In Vitro Antifungal Susceptibility Testing

Microbroth susceptibility testing for *R. arrhizus* and *L. corymbifera* sporangiospores and hyphae was performed against L-AmB (AmBisome^®^, Gilead Sciences Inc., Foster City, CA, USA), AmB (Acros Organics/Thermo Scientific, Waltham, MA, USA), and posaconazole (Thermo Scientific, Waltham, MA, USA) as described in the CLSI document M38-A [38]. Ninety-six-well plates with a flat bottom were used. Concentrations of amphotericin B, liposomal amphotericin B, and posaconazole were added with sporangiospores or hyphae (1.0 × 10^4^–5.0 × 10^4^ sporangiospores or hyphae/mL) in a selective RPMI liquid medium mentioned above in the plates and incubated at 37 °C. The minimum inhibitory concentration (MIC, µg/mL) values were read visually after 24 and 48 h of incubation and determined using the endpoint of the complete inhibition of growth for all antifungals. Experiments were conducted in octuplicate and replicated.

### 2.4. Galleria mellonella Infection and Treatments

Larvae of *Galleria mellonella*, the greater wax moth belonging to the order Lepidoptera, family Pyralidae, were obtained from commercial vendors (Vanderhorst, Inc., St. Marys, OH, USA, and Speedy Worm, Alexandria, MN, USA) in their final larval stage, and stored in wood shavings in the dark at 4 °C. Larvae weighing 250 ± 50 mg were placed in sterile petri dishes containing a sheet of Whatman filter paper. Inoculum suspensions of sporangiospores and hyphae of *R. arrhizus* and *L. corymbifera* ranging from 10^1^ to 10^6^ sporangiospores or germinated sporangiospores per worm were to infect the larval hemocoel via the distal left proleg (10 µL/larva) using the dispenser (PB600-1 Dispenser (Cat No: 83700)) with a Hamilton syringe (500 µL, Model 1750 RN SYR (Cat No: 7658-01)); needle (26 gauge, Large Hub RN needle, 2 in, point style 4, 6/PK, Standard 12° angle beveled needle (Cat No: 7806-03) and incubated at 37 °C for up to 5 days. Survival was recorded daily. For antifungal drug testing, infected larvae were treated with L-AmB 1–2 h post-infection with 5 mg/kg and 10 mg/kg body weight of larva (equivalent to the therapeutic dosages) by injecting on the opposite side (distal right proleg) of the infection side of larvae. All experiments included control larvae injected with sterile 0.9% saline (10 μL/larva) and/or untouched larvae. Serial dilutions of inocula were seeded in RPMI agar plates to confirm the viable counts and calculated based on the number of CFUs on RPMI agar plates incubated overnight at 30 °C.

### 2.5. Statistical Analysis

Data were expressed as means ± standard deviation of the mean. GraphPad Prism 9.5.1.733 software was used to generate and analyze survival curves. Kaplan–Meier survival analysis with the log ranks was used to compare the time of death for each in the infection and treatment group of larvae. We calculated *p* values using the log-rank (Mantel–Cox) test and One-way ANOVA with Dunnett’s multiple comparisons tests and *p* values of <0.05 were considered significant. To calculate the LD_50_ of hyphae and spores, a two-parameter Weibull model was used, from which LD_50_ values were determined by using NLIN, SAS version 9.4. To compare the LD_50_ of spores and hyphae from each species, estimated ratios of the effect doses and the Wald test were used to calculate the significance.

## 3. Results

### 3.1. Comparative Virulence of R. arrhizus and L. corymbifera Hyphae and Sporangiospores Infections in G. mellonella

Previous studies demonstrated that some strains of *R. arrhizus* and *L. corymbifera* caused at least 30% lethality in *G. mellonella* when 10^4^ sporangiospores were utilized [29]. To compare the lethality of hyphae and sporangiospores of *R. arrhizus* ATCC 56536 and *L. corymbifera* ATCC 46771, we employed inocula ranging from approximately 10^1^ to 10^6^ CFUs per larva.

For *R. arrhizus*, larvae were infected with sporangiospores inocula ranging from 2.25 × 10^1^ to 1.36 × 10^6^ CFUs per larva, and an equal number of larvae were infected with hyphae inocula ranging from 2.5 × 10^1^ to 1.25 × 10^6^ CFUs per larva. Hyphal inocula resulted in a lower survival of larvae compared to sporangiospores (Figure 1A–C). At Day 1 (24 h post-infection), all but the highest hyphal infectious doses showed higher mortality compared to sporangiospores (Figure 1C). By Day 5, three hyphae-infected groups contained surviving larvae, whereas four sporangiospores-infected groups contained surviving larvae. Of those groups containing survivors by Day 5, hyphal-infected larvae had a 10% (2.68 × 10^3^ group), 60% (1.34 × 10^2^ group), and 100% (2.50 × 10^1^ group) survival rate, respectively, while sporangiospores-infected groups had 25% (1.88 × 10^4^ group), 60% (1.85 × 10^3^ group), 90% (1.45 × 10^2^ group), and 100% (2.25 × 10^1^ group) survival, respectively (Figure 1A,B). Dose-response curves were generated, and the LD_50_ values for *R. arrhizus* hyphae and sporangiospores were estimated to be 2.02 × 10^2^ and 1.93 × 10^3^ per larva, respectively (Table 1), showing an approximately 10-fold increase in hyphal virulence; these LD_50_ values were significantly different as measured by the estimated ratio of dose effects (*p* = 0.03). These findings demonstrate that *R. arrhizus* hyphae are more virulent than sporangiospores in the *G. mellonella* model.

For *L. corymbifera*, larvae were infected with inocula of sporangiospores ranging from 3.0 × 10^1^ to 2.18 × 10^6^ CFUs, and an equal number of larvae were infected with hyphae using inocula ranging from 2.5 × 10^1^ to 1.78 × 10^6^ CFUs per larva. As with *R. arrhizus* infections, we observed that *L. corymbifera* hyphae needed less inoculum and time to kill *G. mellonella* than sporangiospores (Figure 2A–C). On Day 1 (24 h post-infection), the second- and third-highest inocula of sporangiospores (2.83 × 10^5^ and 2.18 × 10^6^) and hyphae (1.78 × 10^6^, 2.38 × 10^5^, or 1.55 × 10^4^) induced mortality in *G. mellonella* (Figure 2A,C). The observation of hyphae-inducing lower larvae survival was consistent across lower inocula and 5 days of observation (Figure 2A,B). As with the previously described *R. arrhizus* infections, fitted dose-response curves were generated using a two-parameter Weibull model with survival data. The LD_50_ values for *L. corymbifera* hyphae and sporangiospores were estimated to be 3.78 × 10^1^ and 6.21 × 10^2^ per larva, respectively, showing a 16-fold increase in hyphal virulence (Table 1). These LD_50_ values were compared using the estimated ratio of dose effects and found to be significantly different (*p* < 0.001). Moreover, the LD_50_ values for sporangiospores and hyphae of *L. corymbifera* were approximately 3 and 5 times lower than the LD_50_ values for sporangiospores and hyphae of *R. arrhizus*, respectively. Thus, these findings also demonstrate that hyphae of *L. corymbifera* are also more virulent than sporangiospores in the *G. mellonella* model of mucormycosis.

### 3.2. Efficacy of Liposomal amphotericin B against Hyphae and Sporangiospores of R. arrhizus and L. corymbifera in G. mellonella

To further validate the *R. arrhizus* and *L. corymbifera* infectious models in *G. mellonella*, we evaluated the efficacy of L–AmB, a standard of care treatment of mucormycosis in humans. Larvae were infected with a minimum inoculum dose of sporangiospores or hyphae calculated to cause 90-to-100% mortality. Larvae of *G. mellonella* infected with hyphae or sporangiospores were treated with two dosages of L–AmB (5 mg/kg and 10 mg/kg) previously reported to successfully treat *Rhizopus* spp., which caused invasive pulmonary mucormycosis in mice [39].

In *R. arrhizus* sporangiospores infections, we observed almost no difference between L–AmB treatment groups and the infection control group treated with saline on Day 1; the survival rate was 73.4% and 93.4% when treated with L–AmB 5 mg/kg and 10 mg/kg, respectively, compared to 90% larvae survival of infection control group treated with saline (Figure 3A,D). The efficacy of L–AmB (10 mg/kg) against sporangiospores infections increased over time, with a significantly higher survival rate of 53.33% compared to infection groups that were treated with saline and L–AmB (5 mg/kg) on Day 5: 10% and 13.33% survival, respectively. The difference between L–AmB (10 mg/kg) treatment and saline treatment of sporangiospores was statistically significant by One-way ANOVA with Dunnett’s multiple comparison test *p* < 0.001 and Log-rank (Mantel–Cox) test: *p* = 0.023. By comparison, L–AmB treatment failed to produce any therapeutic effect against hyphae as all larvae treated with L–AmB dosages succumbed to the infection on Day 1 (Figure 3B,D).

The survival rate of *L. corymbifera* sporangiospores infections treated with L–AmB was higher than the saline-treated control on Day 1 post-infection, with 87.5% and 95% survival for 5 mg/kg and 10 mg/kg of L–AmB, respectively, compared to 77.5% of infected larvae that survived when treated with saline (Figure 4A,D). This trend remained throughout the experiment, with a higher survival rate of drug-treated larvae compared to saline-treated larvae group that was infected with sporangiospores on Day 5: L–AmB 10 mg/kg and L–AmB 5 mg/kg treatments resulted in 50% and 25% survival, respectively, compared to 12.5% survival of the saline treatment group (Figure 4A,C). The difference between L–AmB (10 mg/kg) treatment and saline treatment of sporangiospores was statistically significant by one-way ANOVA with Dunnett’s multiple comparison test *p* < 0.001 and Log-rank (Mantel–Cox) test: *p* < 0.001. By comparison, all larvae infected with *L. corymbifera* hyphae succumbed to disease within one day regardless of L–AmB dose (Figure 4B,D).

In all the experiments, the control groups of uninfected larvae that received only saline or L–AmB at dosages of 5 mg/kg or 10 mg/kg showed 100% survival (data not shown). These data confirm that the toxicity of either the drug or the saline carrier did not confound the observed results of L–AmB treatment against sporangiospores and hyphae infections. The results suggest that L–AmB at a 10 mg/kg dosage is effective against *R. arrhizus* and *L. corymbifera* sporangiospore infections in *G. mellonella* but not against hyphal infections.

### 3.3. In Vitro Susceptibility of Antifungal Agents against R. arrhizus and L. corymbifera

Due to the lack of efficacy of L–AmB against hyphal mucormycosis in *G. mellonella*, we investigated whether hyphae were more resistant to antifungal agents in vitro. In addition to L–AmB, we tested other standard-of-care antifungal agents for mucormycosis: the original formulation of amphotericin B (AmB) and posaconazole.

For *R. arrhizus* sporangiospores, after 24 and 48 h of incubation, the MIC of all three antifungal agents was 2 µg/mL (Table 2). After 48 h of incubation of hyphae with posaconazole, AmB, or L–AmB, the MICs were 8 µg/mL, 8 µg/mL, and greater than 16 µg/mL, respectively (Table 2), indicating that hyphae are less susceptible to antifungal agents. For *L. corymbifera*, after 24 and 48 h of incubation, the MICs of AmB, posaconazole, and L–AmB against sporangiospores were 2 µg/mL, 4 µg/mL, and 4 µg/mL, respectively. Hyphae were more resistant to all three antifungal agents with MICs of 8 µg/mL after 24 h and MICs greater than 16 µg/mL after 48 h of incubation (Table 2). Thus, hyphae of both *R. arrhizus* (ATCC 56536) and *L. corymbifera* (ATCC 46771) were less susceptible to AmB, L–AmB, and posaconazole than sporangiospores under the in vitro conditions tested.

## 4. Discussion

*Mucorales* species are ubiquitous filamentous fungi found in soil and decaying organic matter [40,41]. *Mucorales* spp., including *R. arrhizus* and *L. corymbifera*, have been shown to form hyphal biofilms under in vitro conditions [42]. Although we are unaware of any reports on *Mucorales* biofilms in nature, it is plausible that such biofilms exist in various environments. For example, plant pathogenic molds readily form biofilm on various plant surfaces [43,44]. When filamentous fungi develop a biofilm, they are less susceptible to harsh conditions, including heat, cold, ultraviolet light, and three fungicides, than planktonic counterparts [45]. Thus, in addition to sporangiospores, hyphal biofilms can be an environmental source for maintaining and spreading filamentous fungi. Soil-borne Mucorales biofilms have also been proposed to be a causative agent during post-traumatic mucormycosis [15]. *Apophysomyces* spp. and *Saksenaea* spp. are extremely rare causes of respiratory infections but are the leading cause of primary cutaneous infections worldwide. In fact, these two species caused two-thirds of all angio-invasive fungal infections in U.S. casualties in Afghanistan [46]. Notably, both species are very poor sporangiospore producers [27], which we observed during our attempts to sporulate *Saksenaea* (data not shown). This may implicate that wound mucormycosis is caused by hyphal biofilms of *Apophysomyces* spp. and *Saksenaea* spp. rather than sporangiospores. Future studies should investigate the prevalence of Mucorales biofilms in nature and their role in trauma-inoculated mucormycosis.

This is the first report to our knowledge to compare the virulence of sporangiospores and hyphae of species from the order Mucorales. Antifungal innate immunity in humans is largely driven by neutrophils, which can destroy hyphae via oxidative burst and nutritional immunity [47,48,49,50,51]. Neutrophils also control fungal infection via neutrophil extracellular traps (NETs), which entrap and expose extracellular fungi to antimicrobial peptides [52]. *G. mellonella* lacks adaptive immunity but mounts an innate immune and humoral response to infection that shares similarities with mammalian innate and humoral immunity [53,54,55]. *Galleria* immune cells, such as plasmacytes and granulocytes, exert cellular immune response by phagocytosis, nodulation, and encapsulation. Additionally, the humoral response, when activated, produces antimicrobial peptides and reactive oxygen/nitrogen species. In our study, *Mucorales* hyphae caused 10-to-16 times more larvae mortality than sporangiospores. Hyphae are likely more resistant to engulfment by phagocytes than sporangiospores since hyphae are generally much larger. The greater virulence of hyphae may also be understood by increased biomass in relation to innate host defense mechanisms. Antachopoulos and colleagues found that fungal biomass is a key factor affecting polymorphonuclear leukocyte-induced hyphal damage of filamentous fungi [56]. Hyphal damage decreased with increasing biomass following the sigmoid (E_max_) model (median R^2^ = 0.87). Moreover, the metabolic activity of different species of Mucorales increases as a sigmoidal function as hyphal biomass increases [57]. These factors also would likely contribute to the increased virulence against innate host defense mechanisms of *G. mellonella* and in vivo resistance to antifungal agents observed with hyphae of *R. arrhizus* and *L. corymbifera* in this study. They also may contribute to devastating tissue injury and resistance to antifungal therapy observed clinically in trauma-induced wound mucormycosis. Thus, limited *G. mellonella* immune responses may be ineffective against hyphae. Similarly, since severe traumatic mucormycosis is also intrinsically resistant to resolution and treatment in human cases, our studies may support those primary innate antifungal responses are likely insufficient to resolve infection regardless of the organism; these questions should be investigated in future studies in higher organisms.

Minimum inhibitory concentrations (MICs) are commonly used to determine the in vitro susceptibility of invasive molds to various antifungal drugs. Notably, while several reports showed no differences in MICs between sporangiospores and germinated sporangiospores in *A. fumigatus*, several studies identified differences in the susceptibility of sporangiospores of *Mucorales* species. AmB and posaconazole were reported to inhibit the metabolic activity of sporangiospores of *L. corymbifera* and *R. arrhizus* strains to a greater degree than that of hyphae [58]. Labuda et al. tested the MIC of polyenes, azoles, and echinocandins with hyphae (mycelial microcolonies) of *Saksenaea dorisiae* and found that hyphae were more resistant to antifungals compared to sporangiospores [59]. In fact, hyphae were resistant to all antifungals tested except terbinafine. These data also suggest that AmB and posaconazole may not be the optimal treatment against *S. dorisiae* hyphae. Likewise, our MIC data indicate that hyphae of *L. corymbifera* and *R. arrhizus* strains are less susceptible to AmB than sporangiospores. Since drugs target growing cells, one explanation may be that cells are killed at the germination stage when germlings actively grow and are most vulnerable. The differences in in vitro susceptibility between hyphae and sporangiospores underscore the importance of establishing standardized protocols and guidelines for performing MIC assays to accurately determine the susceptibility of *Mucorales* species to different antifungal drugs.

While our study offers valuable insights into the virulence of various forms of *R. arrhizus* and *L. corymbifera*, as well as their response to antifungal agents, we acknowledge several limitations of our study. Although the study compares the virulence of different fungal forms, it falls short of fully elucidating the mechanism of death and progression of infection in *Galleria* larvae. It is also important to acknowledge that further studies in pre-clinical animal models are necessary before the data can be considered directly applicable to clinical decision-making regarding appropriate dosages of antifungal agents for patients.

## Figures and Tables

**Figure 1 jof-09-00958-f001:**
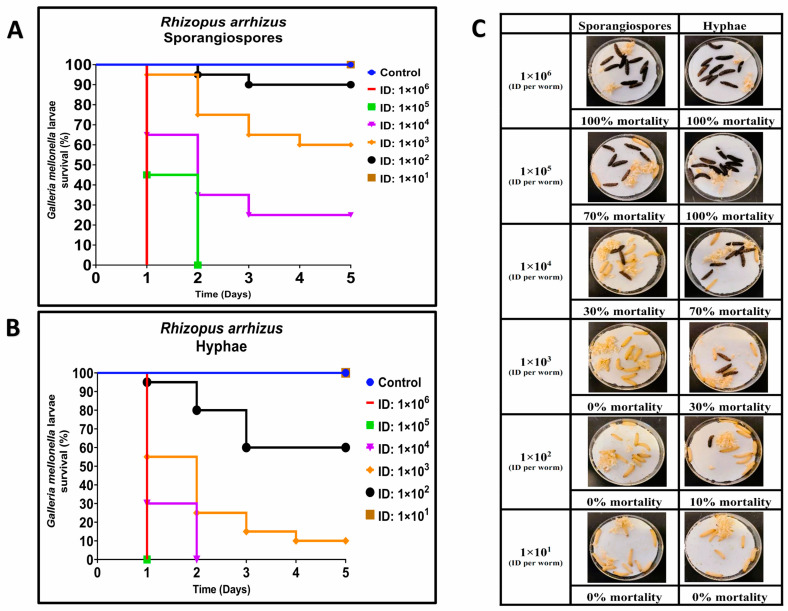
Comparative virulence of *Rhizopus arrhizus* (ATCC 56536) sporangiospores and hyphae infections in *G. mellonella*; (**A**,**B**): Day-wise dose-dependent survival of *G. mellonella* after *R. arrhizus* sporangiospores and hyphae infection; (**C**): Comparative survival of worm images between sporangiospores and hyphae observed 24 h post infections. (Note: The number of larvae in Figure 1C represents the comparison of sporangiospores and hyphae infection of a single experiment; the appearance of black color larva = dead; the appearance of beige color larva = alive; ID: inoculum dose).

**Figure 2 jof-09-00958-f002:**
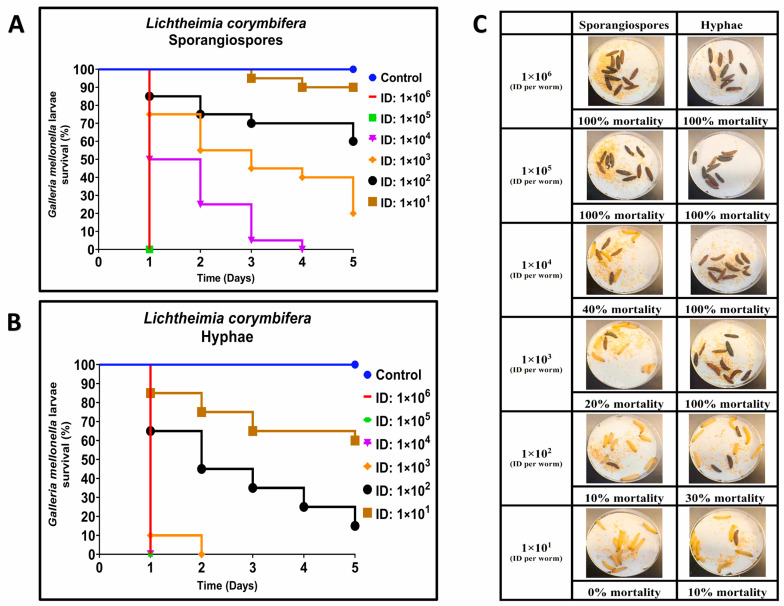
Comparative virulence of *Lichtheimia corymbifera* (ATCC 46771) sporangiospores and hyphae infections in *G. mellonella*; (**A**,**B**): Day-wise dose-dependent survival of *G. mellonella* after *L. corymbifera* sporangiospores and hyphae infection; (**C**): Comparative survival of worm images between sporangiospores and hyphae observed 24 h post infections. (Note: The number of larvae in Figure 2C represents the comparison of sporangiospores and hyphae infection of a single experiment; the appearance of black color larva = dead; the appearance of beige color larva = alive; ID: inoculum dose).

**Figure 3 jof-09-00958-f003:**
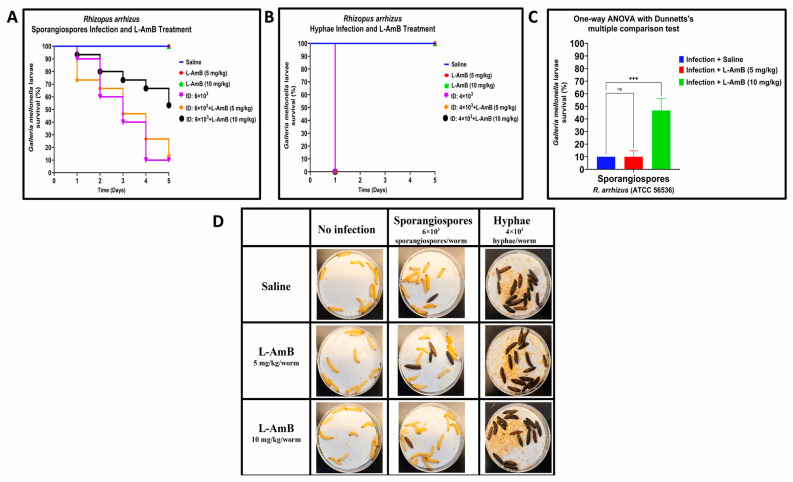
Comparative efficacy of liposomal amphotericin B (L–AmB) treatment on *Rhizopus arrhizus* (ATCC 56536) sporangiospores and hyphae infections in *G. mellonella*; (**A**,**B**): Day-wise dose-dependent survival of *G. mellonella* after L–AmB treatment on sporangiospores and hyphae infection; (**C**): Statistical analysis of survival of larva after 5 days post-treatment; (**D**): Survival images of larva comparing the 24 h post-L–AmB treatment of sporangiospores and hyphae infections. (Note: The number of larvae in Figure 3D represents the comparison of sporangiospores and hyphae infection of a single experiment; the appearance of black color larva = dead; the appearance of beige color larva = alive; statistical significance: ***: *p* < 0.001, ns: nonsignificant; ID: inoculum dose).

**Figure 4 jof-09-00958-f004:**
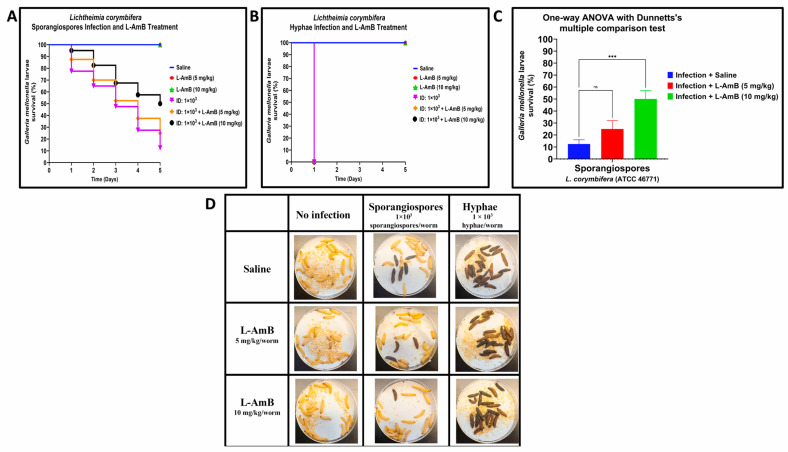
Comparative efficacy of liposomal amphotericin B (L–AmB) treatment on *Lichtheimia corymbifera* (ATCC 46771) sporangiospores and hyphae infections in *G. mellonella*; (**A**,**B**): Day-wise dose-dependent survival of *G. mellonella* after L–AmB treatment on sporangiospores and hyphae infections; (**C**): Statistical analysis of survival of larva after 5 days post-L–AmB treatment; (**D**): Survival images of larva comparing the 24 h post-L–AmB treatment of sporangiospores and hyphae infections. (Note: The number of larvae in Figure 4D represents the comparison of sporangiospores and hyphae infection of a single experiment; the appearance of black color larva = dead; the appearance of beige color larva = alive; statistical significance: ***: *p* < 0.001, ns: nonsignificant; ID: inoculum dose).

**Table 1 jof-09-00958-t001:** Virulence of *R. arrhizus* and *L. corymbifera* in *G. mellonella*.

Strain	LD_50_	
	Sporangiospores	Hyphae
***R. arrhizus* (ATCC 56536)**	1.93 × 10^3^ ± 7.38 × 10^2^ *	2.02 × 10^2^ ± 5.40 × 10^1^ *
***L. corymbifera* (ATCC 46771)**	6.21 × 10^2^ ± 2.57 × 10^2^ **	3.78 × 10^1^ ± 1.59 × 10^1^ **

**Note:** * *p* = 0.03; ** *p* < 0.001.

**Table 2 jof-09-00958-t002:** In vitro susceptibility of sporangiospores and hyphae of *Rhizopus arrhizus* (ATCC 56536) and *Lichtheimia corymbifera* (ATCC 46771) against posaconazole, amphotericin B, and liposomal amphotericin B (µg/mL) at 24 and 48 h of incubation.

Minimum Inhibitory Concentration (MIC)		Posaconazole (µg/mL)		Amphotericin B (µg/mL)		L-Amphotericin B (µg/mL)	
		24 (h)	48 (h)	24 (h)	48 (h)	24 (h)	48 (h)
** *Rhizopus arrhizus* ** **(ATCC 56536)**	**Sporangiospores**	2	2	2	2	2	2
	**Hyphae**	4	8	4	8	8	>16
***Lichtheimia corymbifera* (ATCC 46771)**	**Sporangiospores**	4	4	2	2	4	4
	**Hyphae**	8	>16	8	>16	8	>16

## Data Availability

The data are available upon request from the corresponding author.

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
