# Peer review of "Hyphae of Rhizopus arrhizus and Lichtheimia corymbifera Are More Virulent and Resistant to Antifungal Agents Than Sporangiospores In Vitro and in Galleria mellonella"

_jof, 2023, doi:10.3390/jof9100958_

Round 1
Reviewer 1 Report
The study is very interesting, well conducted and expands our vision to understand that in fungal infections, depending on how they are acquired, they can assume different virulence.
I see this article as an initial study on the subject. And sometimes the authors are very categorical in generalizing the results.
The Galleria mellonella model is elegant. However, it is necessary to expose the limitations that can be made with the human being. Likewise, the number of isolates tested and therefore the results of in vitro susceptibility tests are limited. More studies are needed to establish these findings. I'm not sure, as written in the article, "that the data would be useful to clinicians in selecting appropriate dosages of antifungal agents for patients with R. arrhizus and L. corymbifera infections. In the same vein, the in vitro sensitivity of terbinafine should be viewed with caution.
Author Response
Response to Reviewer 1:
We appreciate the reviewer's thoughtful comments and suggestions. We acknowledge that our study provides initial insights into the different virulence of hyphal and sporangiospore forms of Mucorales species.
- We agree that our results should be interpreted cautiously and that further studies are needed to establish and generalize these findings. We have included the limitations of our study in the revised manuscript, particularly regarding the applicability of our findings to humans. Additionally, we have modified the statement regarding the usefulness of the data for clinicians in selecting appropriate dosages of antifungal agents, acknowledging that further studies are necessary for clinical applications. We have also emphasized the preliminary nature of our results and the need for additional research to confirm and extend our observations.
- Regarding the concern about the in vitro sensitivity of terbinafine, we would like to clarify that our study did not investigate the sensitivity of terbinafine. In our discussion, we aim to emphasize the relevance of the study conducted by Labuda et al. in 2019 (Reference no: 50), which compared minimum inhibitory concentrations of Saksenaea dorisiae hyphae and sporangiospores to different antifungals agents.
Reviewer 2 Report
In this manuscript the authors identify and describe an improved model to test the potential virulence and antifungal susceptibility to two Mucorales species. While it was generally believe that the infectious inoculum of Mucorales are sporangiospores, the authors suggest that hyphal elements are the more likely infectious unit for some infections, particularly combat wound associated infections. These, too, cause significant morbidity. Therefore they compared sporangiospores and hyphal elements in a Galleria survival model. They also tested efficacy of L-Amp B in vivo and several antifungals in vitro. The goal was to describe a better model for wound mucormycosis.
The study is well rationalized and clearly presented. The authors clearly describe methods for fungal preparation and inoculation. The results are convincing that hyphal elements are more virulent in a Galleria model – about 10 fold more. They speculate that this may be because the cellular response is less efficient at killing hyphae vs sporangiospores, possibly due to increased biomass. This is a rational conclusion.
Are the Galleria just dying from blockage?
Do hyphae grow in the larvae?
Do sporangiophores proliferate?
The authors also tested the efficacy of L-AmpB in the in vivo model. The antifungal was less efficient in both species at higher inoculums of hyphal elements vs sporangiospores. This observation was confirmed when determining MICs. There was always a trend toward and/or increased MIC with hyphal vs sporangiospores.
Consequently, the authors present an improved model to test antifungal sensitivity to antifungals with regard to combat wound mucormycosis. It is not clear that this model tests the actual virulence of these forms because there is no information on what happens after the initial colonization.
In addition:
a) At what point was antifungal administered?
b) In the graphs it would be helpful to put colored squares to match the key for all inoculations – for instance in figure 1B – a green square around the red square like was done for the brown around the blue
Author Response
Response to Reviewer 2:
We appreciate the reviewer's positive assessment of our study and their valuable comments. We have addressed the questions raised in the review to provide clarity on the experimental procedures and findings:
- While our study focused on comparing the virulence of hyphal and sporangiospore forms, we acknowledge that the mechanism of death in the larvae was not fully elucidated. Also, we did not precisely track the growth of hyphae or proliferation of sporangiospores within the larvae. We have included a discussion of this limitation in the revised manuscript.
- The timing of antifungal administration is an important point. We have included additional information in the Methods section to specify that infected larvae were treated with antifungal agents 1 to 2 hours post-infection.
- We have revised all the figures to enhance the visual representation of the data that match the key for all inoculations for readers to interpret the graphs accurately. We thank you for this valuable feedback.